# Healthcare Professionals’ Perceptions of Loneliness amongst Older Adults: A Qualitative Study

**DOI:** 10.3390/ijerph182212071

**Published:** 2021-11-17

**Authors:** Iria Dobarrio-Sanz, Crístofer Ruiz-González, Cayetano Fernández-Sola, Pablo Roman, José Granero-Molina, Jose Manuel Hernández-Padilla

**Affiliations:** 1Department of Nursing, Physiotherapy and Medicine, University of Almeria, 04120 Almería, Spain; ids135@ual.es (I.D.-S.); cfernan@ual.es (C.F.-S.); prl690@ual.es (P.R.); jgranero@ual.es (J.G.-M.); j.hernandez-padilla@ual.es (J.M.H.-P.); 2Obstetrics & Gynecology Ward, Torrecárdenas University Hospital, 04009 Almería, Spain; 3Facultad de Ciencias de la Salud, Universidad Autónoma de Chile, Santiago 7500000, Chile

**Keywords:** descriptive study, healthcare, loneliness, older adults, public health, qualitative study

## Abstract

Background: Loneliness amongst older adults is linked to poor health outcomes and constitutes a public health issue worldwide. Healthcare professionals’ perceptions could influence the strategies they implement in order to prevent, detect and manage loneliness amongst older adults. The aim of this study was to describe and understand healthcare professionals’ perceptions of loneliness amongst older adults. Methods: A descriptive qualitative study. Twenty-six Spanish healthcare professionals with experience caring for older adults participated in the study. Data were collected between November 2019 and September 2020 using focus groups and in-depth interviews. Data were analysed following a content analysis method using ATLAS.ti software. Results: Healthcare professionals’ perceptions of loneliness amongst older adults is represented by three themes: (1) “when one’s personal life and social context lead to loneliness”; (2) “from abandonment to personal growth: the two faces of loneliness”; and (3) “loneliness as a health issue that needs to be addressed”. Conclusions: Healthcare professionals perceive loneliness as a multifactorial, subjective experience that can trigger different coping mechanisms and negatively affect older people’s health. Healthcare professionals consider that a greater involvement of the whole society is needed in order to fight loneliness amongst older adults as a public health issue.

## 1. Introduction

Loneliness amongst older adults living in the community and in long-term care settings is a public health issue worldwide [1,2,3]. Loneliness can be defined as a state of mind caused by one’s perception of deficient interpersonal relationships [4,5]. Factors such as physical deterioration, emotional losses and social isolation increase the risk of loneliness amongst older adults [6,7]. Globally, the prevalence of loneliness amongst older adults raises up to 40% [8,9]. In Spain, more than two million older adults live alone [10], and almost 20–25% feel lonely [11].

In older adults, loneliness is linked to cardiovascular disease [12,13], decreased levels of physical activity, sarcopenia [8,14,15], hypercholesterolemia, diabetes [16,17], altered inflammatory response [14,18], anorexia [19] and obesity [20]. Loneliness in older people has also been associated with depression [9,21,22], sleep disorders [23], apathy [8] and cognitive decline [24]. In fact, loneliness in older people can be considered as a health risk factor equivalent to smoking, alcohol consumption or sedentary lifestyles [13,14]. In this regard, loneliness amongst older people is not only associated with longer hospital admissions and a higher rate of hospital readmission [9,25] but also with a higher absolute mortality rate [14,26].

Despite the significant impact of loneliness on older adults, the difficulty in identifying this problem remains an obstacle for healthcare professionals to take measures [27,28]. In relation to this, different studies have tried to find out the role of loneliness in both institutionalised and community-dwelling older adults [7,29,30,31,32,33,34], comprehend healthcare professionals’ experience on older adults’ isolation [28] and evaluate strategies to reduce loneliness in older adults [35,36]. However, more research focusing on understanding healthcare professionals’ perceptions of the phenomenon of loneliness in older adults is needed [28,37,38] since evidence suggests that healthcare professionals receive insufficient training to address this important issue [37,39,40]. Understanding how healthcare professionals depict and perceive loneliness amongst older adults would allow us to approach the problem from another angle so that we can develop interventions and preventive strategies that resonate with the healthcare professionals who will implement them [28,37,41]. This could also make older adults feel better understood by healthcare professionals [35]. The objective of this study was to describe and understand the perceptions that health professionals have about loneliness in older adults.

## 2. Materials and Methods

### 2.1. Design

A descriptive qualitative study was carried out [42]. This approach is based on the principles of naturalistic inquiry and allow researchers to describe phenomena focusing on how the participants view, interpret or live a given phenomenon in its natural state [42,43]. We followed the Consolidated Criteria for Reporting Qualitative Research (COREQ) to write the present manuscript [44].

### 2.2. Participants and Setting

The participants were healthcare professionals recruited using convenience sampling methods. The participants met the following inclusion criteria: (1) having at least 6 months of professional experience looking after older adults and (2) signing an informed consent form before participating. The study sample comprised 26 white European healthcare professionals, whose average age was 28.6 years old and had an average professional experience of 6 years looking after older people in different settings (Table 1). The study was carried out in the southeast of Spain between November 2019 and March 2020.

### 2.3. Data Collection

Data were collected between November 2019 and March 2020 using focus groups (FGs) and in-depth interviews (IDIs). FGs and IDIs were conducted following an interview protocol developed by the research team (Table 2). The FGs and IDs were conducted by 2 researchers with ample experience in data collection techniques in qualitative research. The two FGs were conducted as a first step in the exploratory phase of the study. One of the FGs comprised different healthcare professionals with experience caring for older adults in a variety of contexts (interdisciplinary focus group—IFG), whilst the other FG comprised nurses with experience looking after older populations. The FGs and the 14 IDIs were carried out in a private room within the university where the researchers work. Both FGs lasted 55 and 62 min, whilst the IDIs lasted 40–55 min (average = 46 min). Both FGs and IDIs were audio-recorded and transcribed verbatim in Spanish. The interview scripts and the most relevant quotes from participants were translated by a professional native English translator for inclusion in this manuscript. The transcriptions and the fieldnotes taken by the researchers were incorporated into a hermeneutic unit and analysed using ATLAS.ti (ATLAS.ti Scientific Software Development GmbH version 8). Data collection was ceased when the IDIs stopped providing new information (data saturation reached).

### 2.4. Data Analysis

Data were analysed following a thematic analysis method [45,46]: (1) Data familiarisation: after transcribing the data, researchers read all the transcriptions to gain a general understanding of what participants said and then reread them to add familiarisation comments using the “add memo” function in ATLAS.ti. (2) Systematic data coding: researchers selected significant quotes and assigned them codes using the function “open coding” and “coding in vivo” in ATLAS.ti. (3) Generating initial themes from coded and collated data: researchers generated initial themes by grouping codes that shared meaning patterns and were related through a central idea. (4) Developing and reviewing themes: researchers double-checked that all the generated themes were consistent with the codes they included and the quotes upon which they were developed. (5) Refining, defining and naming themes: researchers reviewed the final three themes and refined them by merging some of them. At this point, the definitive names of the themes were created. (6) Writing the report: when preparing this research report, the researchers selected the most illustrative quotes and summarized the explanation of each theme/subtheme. Then, the researchers refined the report by filtering the absolutely essential fragments and relating them to the research question and the literature review.

### 2.5. Rigor

Methodological research was assured at every stage of the study following Lincoln and Cuba’s [43] quality criteria. Credibility: the data collection process was described in detail, and its interpretation was supported by researcher triangulation. Finally, the analytical process was revised by two independent reviewers. Transferability: the method, participants, setting and context of the study were described in detail. Reliability: two researchers, who did not participate in data collection and had expertise in qualitative research and elderly people’s care, corroborated the analysis. Confirmability: the triangulation in the analysis was subsequently validated by the study’s healthcare workers, who confirmed the accuracy and interpretation of the transcriptions.

### 2.6. Ethical Issues

The study was carried out following the ethical principles of the Declaration of Helsinki. Consent was obtained by the Ethics and Research Committee (EFM892020). All of the participants were aware of the study’s objective, the voluntary nature of their participation and the possibility to stop at any point. Confidentiality and anonymity were guaranteed in compliance with the Organic Law 3/2018 of 5 December on Data Protection and Guarantee of Digital Rights. Digital rights were guaranteed in line with the European Parliament’s Regulation 2016/679. Informed consent was obtained from all participants before starting data collection.

## 3. Results

Three main themes were extracted from the data analysis. These three themes characterize healthcare professionals’ perceptions of loneliness amongst older people (Table 3).

### 3.1. When One’s Personal Life and Social Context Lead to Loneliness

The theme represents loneliness as a reality favoured by different factors. Loneliness amongst older adults could be associated with internal triggers such as the individual’s health, external triggers such as the individual’s economic situation or a combination of both. This theme comprises two subthemes.

#### 3.1.1. Age, Physical Condition and Character Traits: Individual Obstacles Limiting Accompaniment

Loneliness is perceived to be conditioned by how older people interpret and feel about their health. These interpretations and feelings could condition their priorities in life and influence their decision-making process regarding whether or not to live alone. Similarly, healthcare professionals perceive that other health-related factors limiting older people’s autonomy could influence the degree to which they feel lonely.

Participants insisted that loneliness amongst older people is sometimes imposed by a combination of disease-related limitations (e.g., impaired mobility) and architectural barriers at home, which in conjunction can prevent older people from going out and interacting with their community.
*They might have had a stroke and be in a wheelchair and have mobility problems, so their home and the means they have at home are not adequate for them to be able to interact with their neighbours.*(FG-1) ID7

Healthcare professionals perceived that living alone could be a risk factor for older adults to feel lonely. However, our participants pointed out that sometimes older people prefer to live alone. The decision to live alone is made on the basis of the possibilities, limitations and constraints that older people encounter. Participants reported cases of older people who prefer to be alone than to have an unchosen companion in a long-term care centre for older adults.
*They prefer to be at home alone, even if they do not talk to anyone, because of the experiences they have heard from others, and they do not want to try socialising in other centres.*(FG-3) ID8

Loneliness among older adults has also been attributed by some participants to cultural barriers and individuals’ character traits such as being an introvert or being fearful of engaging with strangers.
*I think that culturally, they don’t want to go to places, residences or day hospitals because they are afraid of the unknown or their culture doesn’t allow them to do it.*(FG-2). ID6

#### 3.1.2. Social Factors That Push Older Adults towards Loneliness

This subtheme includes external factors that healthcare professionals perceive could contribute to isolation amongst older adults. Most external factors identified by participants are socioeconomic, and they are related to new family models, social attitudes towards older people, geographical isolation or the lack of institutional and/or economic resources.

Job commitments and family responsibilities could influence the way people relate to and live with other people. Busy lives with strict work schedules and numerous obligations could condition the amount of time younger generations spend with older adults, which could lead to loneliness amongst the latter.
*It is also true that many family members cannot take care of their elderly because they have no time and spend most of the day at work*.(ID2)

Older people’s economic situation has also been identified as an issue that may contribute to increased loneliness in older adults. In the absence of family support, some people cannot afford to pay for professional home support services, which not only affects their loneliness but also contributes to further health deterioration.
*They have hired a girl to come in the morning for food and a girl to sleep. But I know people who have neither family support nor money to pay for these expenses. So the economic issue does mean that they are less lonely.*(FG-3) ID9

In Spain, long-term care centres for older adults have high occupancy rates. Social services do not always find vacancies in local areas (i.e., near the older adults’ previous home). Being allocated to geographically distant long-term care centres is perceived to contribute to limiting family visits and, therefore, foster loneliness amongst older adults.
*I knew of a case where (...) the residence was in a village and it was not so easy for the relatives to be go if they were working in the city.*(FG-4) NFG6

### 3.2. From Abandonment to Personal Growth: The Two Faces of Loneliness

Given the individual and unique nature of loneliness, the way older people perceive loneliness can vary from one person to another. On the one hand, some older people interpret being lonely as a consequence of not being a burden on their loved ones nor an obstacle to their professional development. On the other hand, other older people perceive loneliness as a consequence of being abandoned and neglected. This theme comprises two subthemes.

#### 3.2.1. Loneliness as an Opportunity: Between Introspection and Flight

This subtheme narrates how healthcare professionals perceived that loneliness can be experienced as an opportunity when it is related to older adults choosing to live alone in order to avoid becoming a burden on their relatives. Although loneliness is associated by definition to negative feelings, healthcare professionals perceived that many older adults are able to confront the situation with an optimistic approach that helps them to combat it.

Healthcare professionals perceived that for some older adults, living alone in their usual environment can be an opportunity to release tension, relax and regain the energy needed to continue their daily lives with physical and psychological strength.
*He was very nervous and found it difficult to adapt to the hospital room where he was admitted. I remember that there were many family members with him. When I finally spoke to him, he expressed his need to go home, to be alone and to feel his inner calmness.*(FG-4) ID8

Similarly, feeling lonely has been seen as a trigger to regain inspiration for new activities and hobbies that had been displaced by other needs or obligations. Thus, activities such as reading or doing exercise could provide additional health benefits for older adults.
*There are elderly people who, living alone, completely change their lifestyle. They can do activities they have always wanted to do and devote more time to them.*(ID1)

Feeling lonely can also trigger the need for finding ways to escape from reality, leaving behind worries that may have been affecting older adults’ wellbeing. In fact, some healthcare professionals perceived that those older adults who manage to see loneliness as an opportunity to focus on themselves had lower levels of anxiety and paid less attention to external problems.
*In some cases, when they are alone and feeling lonely, they realise that this gives them time for themselves, to move away from the problems that are going on around them and they feel calmer.*(FG-2) ID12

#### 3.2.2. Loneliness as a Source of Negative Feelings

Loneliness does not always mean there is a lack of formal or informal support network for older adults. Lack of motivation, lack of interest to plan activities or the monotony of daily routine are also perceived as causes of loneliness amongst older adults.

Loneliness can uncover a range of emotions that have a negative impact not only on older adults’ health but also on their relatives’ psychological wellbeing. Healthcare professionals linked loneliness to feelings of abandonment and perceived that older adults often interpret their loneliness as the consequence of an act of cruelty from their relatives. Older people’s families and relatives are aware of this interpretation, and they feel guilty.
*They feel lonely, depressed, abandoned by their families.*(FG-2) IFG6
*I think the elderly often feel abandoned and their relatives know it and feel guilty about it. In the end they are all psychologically affected by it.*(FG-2) ID4

In some cases, being surrounded by people does not mean that the feeling of loneliness disappears and older adults feel sheltered or accompanied. Sharing space and time with other older adults in similar conditions (for example: in a long-term care centre) does not eliminate the feeling of emptiness, since for some participants, the true meaning of companionship is to be found amongst people they trust and have close affective relationships with.
*Families believe that they are not alone because they are in a nursing home with company; however, deep down they feel lonely*.(FG-1) NFG5

The feeling of abandonment experienced by some older adults could lead to letting themselves go, to stop doing activities, to stop maintaining an active life or to stop taking care of their own health. Hopeless prospects for the future without the hope of companionship lead to feelings of indifference and disillusionment.
*Loneliness in my case referred to the feeling she had and she didn’t schedule activities and she didn’t feel like doing anything other than being alone.*(FG-3) IFG2

### 3.3. Loneliness as a Health Issue That Needs to Be Addressed

Loneliness has a negative impact on the health of the older adult who experiences it, but it also affects healthcare professionals and the healthcare system of which they are all part. Loneliness amongst older adults is now considered a public health issue that social and healthcare services cannot ignore. Within this theme, two subthemes were generated: “loneliness as a health issue” and “the need for a multidimensional solution”.

#### 3.3.1. Loneliness as a Health Issue

Loneliness has a direct impact on older people’s health. Although there are many indicators that suggest an older adult may be feeling lonely, these are not always easy to identify. Somatisation is a common manifestation in older adults who feel lonely. This refers to a phenomenon that occurs when the patient repeatedly seeks medical help, has symptoms associated with significant distress and impairment and there is no biological evidence of organic disease.
*My grandmother often reflects her feelings in physical things in order to get medical attention and dialogue with health professionals.*(FG-3) ID13

The participants agreed that healthcare professionals try to look for the cause of the patient’s symptoms by focusing on the plausible biophysiological cause and neglecting important aspects such as loneliness or emotional balance. Healthcare professionals perceive that problems with no apparent solution are often caused by feeling lonely or abandoned and not sharing this feeling with anyone.
*When I realised, I left the room without taking his blood pressure because all he wanted to do was talk to you and it is a demand for attention.* ID1

Occasionally, older adults experience rapid and progressive health deterioration, frequently expressing their discomfort. Although the patient himself is aware of the origin of his problem, they are unable to verbalise it, and it is therefore necessary that the people around them know how to identify it.
*She said she wanted to go to the hospital and we didn’t realise that it was a way of somatising in order to be more attentive to her.*(FG-2) NFG3

Healthcare professionals perceive that loneliness amongst older adults lead to self-care problems (i.e., self-neglect, lack of motivation to perform self-care behaviours and failure to attend follow-ups). Participants believe that not having someone close who is aware of the older adult’s condition leads to a situation of vulnerability.
*She didn’t know what time it was, so maybe she ate at 11am and then she didn’t eat again until dinner time, because she had already eaten a meal.*(FG-3) ID10
*They forget the appointment and don’t come. However, those who did have family members, they have their blood tests up to date, and they did not forget their hospital appointments.*(FG-3) ID11

#### 3.3.2. The Need for a Multidimensional Solution

This subtheme refers to the welfare, institutional, human and economic resources that are articulated in an attempt to address loneliness. Firstly, professionals have several limitations that make it difficult to address loneliness: lack of time, knowledge and awareness of the problem.
*Loneliness is not seen as a problem that affects health; there is a lack of knowledge of the health professional both personally and in dealing with these issues.*(FG-1) ID14
*Ideally, we would like to be able to spend more time with them, to spend more time on the emotional aspects, not just on medication and testing.*(FG-1) ID3

The interdisciplinary team should be strengthened, and work should be carried out in a coordinated manner to favour independence and social coexistence. Older adults need to exercise their skills and strengthen their capacities in order to feel useful and to slow down their physical and cognitive decline. In this regard, professionals play an essential role in alleviating the consequences of loneliness in the different spheres of an older adult’s life.
*I think the work of professionals such as the social worker and the physiotherapist is important and my experience was quite good; they did games and activities to delay the cognitive deficit.*(FG-1) IFG4

Often, people who are experiencing loneliness demand healthcare in order to be seen by someone who can identify their problems and needs. In most cases, older adults feeling lonely do not need expensive diagnostic procedures or pharmacological treatments, and they seek help as a means to express their desperate situation, to interact with another person and to feel that they are understood.
*It is not a question of a drug, of a therapy, but of the constancy of being there by the side of a coordinated team of professionals.*(FG-1) ID5

Healthcare professionals perceive that community healthcare resources are very much needed to fight against loneliness amongst older adults. Financial support schemes, easier access to day-care centres and community-care groups could improve loneliness amongst older adults.
*The financial resources provided by the State are a very important help, for example, to have toilets adapted for wheelchairs or people with mobility problems.*(FG-3) IFG6
*Home carers play an important role in the support and well-being of the elderly and make them feel less isolated.*(FG-3) ID4

## 4. Discussion

The aim of this study was to describe and understand healthcare professionals’ perceptions of loneliness amongst older adults. Using a qualitative descriptive design and thematic analysis allowed us to achieve a thorough description of these perceptions from a personal, familial, social and healthcare perspective. The information revolved around three themes: what the healthcare worker participants considered the causes of loneliness, their perceptions of the feelings this situation produces in their patients, and the recognition of loneliness as a healthcare problem where public healthcare workers play a pivotal role in minimising it.

The participants relating the experience of living in solitude to a series of intertwining personal and social factors that determine the way in which people adapt to loneliness over time [47,48]. The aging process is frequently associated with health problems that lead to functional limitations and dependency [49,50]. This results in limited social interactions [51] due to physical consequences such as decreased mobility [52] or due to the architectural impediments of their surroundings [53,54,55]. However, cultural factors or negatives ideas surrounding care homes lead to some older adults preferring being lonely within their own homes than being in the company of people they do not know [56,57]. Nevertheless, the discomfort and the measures taken to fight loneliness are, to a great extent, determined by the decision-making process of the older adults [58,59], hence why some of the participants referred to health conditions or the circumstances of family members as being deciding factors in older adults, leading older adults to suffer imposed loneliness [56,58,60]. This also occurs when their economic resources are limited and they are not able to access professional care to mitigate loneliness [56,58,61], as well as when there is a lack of availability in care homes [62].

The issue is that many older adults face discomfort when they feel that they are a nuisance or a burden on family members, leading them to go to a care home [59,60]. Furthermore, some interpret this decision as a new situation full of potential and a unique opportunity for introspection and to do spiritual activities [7,31,63]. This approach allows for all of society to prepare for a successful aging process and, indirectly, to reduce the feeling of loneliness in older adults [64,65]. Healthcare professionals who participated in this study understood that there was a higher level of loneliness in older adults’ homes than in care homes [29,30], contrary to what other studies have demonstrated [22,32]. Irrespective of the context in which the older adults lived, and even being surrounded by many people, if they did not have the support and company of their loved ones, this generated negative and unpleasant feelings [30,32,66]. This, coupled with boredom and the monotony of daily life, led to feelings of loneliness [24,31,34].

Some of the participants spoke about the fact that many older adults feel depressed, neglected or demotivated [9,22,67], leading to problems in self-care: forgetting to take medication, malnutrition and safety issues [68,69]. The majority emphasize that the somatision is a direct consequence of feeling neglected and lonely [70]. The need to communicate and fill the void of loneliness is evident on a physical level, and it is used as a way to connect with one’s surroundings in an attempt to avoid one’s soul becoming lonely [47,71,72]. Loneliness is self-reinforcing, leading to an even greater sense of isolation [27]. The older adult withdraws himself more and more from the outside world to find internal peace when outer freedom is impossible, and they are overcome with a feeling of hopelessness [27,47].

The participants of this study highlighted the need for an attitude change amongst professional and in society as a whole, as well as an increase in resources to be able to solve the problem of neglect and loneliness felt by older adults [73,74]. They specifically mention a lack of time, planning and personnel [75,76] as well as the need for training and awareness regarding the issue [28,75], which they deem the approach to loneliness. Likewise, healthcare workers must cooperate with each other to provide adequate help and guarantee a multidimensional approach [32,75,77]. Healthcare professionals have been working on different approaches to understand and detect loneliness amongst older adults [30,33,35,41]. In addition, evidence shows that healthcare professionals have implemented different interventions with positive effects on the prevention and management of loneliness amongst older adults [36,78,79,80,81]. Despite this, loneliness is becoming an ever more complex public health problem, and it is necessary to tackle it urgently in a way that involves the whole of society [9,82]. It is not fair to offload the responsibility for loneliness on its protagonists, when they themselves perceive an ever greater distancing from society [33], resulting in a more passive lifestyle, marginalisation and vulnerability [33,83,84].

This study has various limitations that should be considered when interpreting the results. Firstly, this study was carried out in only one geographical area in Spain, which could limit the generalizability of our conclusions. In order to achieve a more representative sample of the target population, various focus groups and interviews were carried out. It would be ideal to include a wider range of scenarios to evaluate the perceptions and experiences of healthcare professionals. In terms of selecting the sample, the study approaches a serious problem from the perspective of healthcare professions who have experience in caring for older adults. However, the perceptions of the older people, themselves, were not taken into account. The way healthcare professionals perceive older adults’ loneliness could differ from the way older adults experience it, themselves. Future studies could address loneliness amongst older adults using a participatory action research approach. Including the perspectives and experiences of both older adults and healthcare professionals could contribute to the development of more robust interventions to fight loneliness amongst this population. Furthermore, the professional experience of older adults varied a great deal amongst the participants; those who had limited experience reduced the transferability of the findings. The large differences amongst the professions of our participants could have influenced our results. Furthermore, recruiting mostly young professionals could have played a part in the way older adults shared their feelings of loneliness, which in turn could have influenced the way the professionals perceived the phenomenon. Regarding data collection, only in-depth interviews and focus groups were carried out. In addition, we used the same protocol for both interviews and focus groups, so we cannot ascertain that the information shared by the participants in the focus groups had not been influenced by the actual group. If the participants had been observed in their work setting, this could have provided additional data.

## 5. Conclusions

Healthcare professionals perceive that loneliness amongst older adults is multifactorial in origin, and there are external social factors and internal personal factors that favour it. Healthcare professionals perceive loneliness as a subjective experience that can trigger different coping mechanisms and negatively affect older people’s health. Healthcare professionals consider that a greater social implication is needed in order to fight the public health issue of loneliness amongst older adults.

## Figures and Tables

**Table 1 ijerph-18-12071-t001:** Sociodemographic data.

Participant	Age	Sex	Professional Experience	Profession	Area	Setting
IDI1	31	F	5 years	Nurse	Nursing home	Rural
IDI2	31	F	4 years	Nurse	Nursing home	Urban, Rural
IDI3	26	F	5 years	Nurse	Nursing home	Urban
IDI4	28	F	6 years	Nurse	Hospital	Urban
IDI5	29	F	6 years	Physician	Primary care	Urban, Rural
IDI6	22	M	6 months	Nurse	Hospital	Rural
IDI7	22	M	6 months	Nurse	Nursing home	Urban
IDI8	34	F	14 years	Nurse	Hospital	Urban, Rural
IDI9	21	F	6 months	Nurse	Primary care	Rural
IDI10	22	F	6 months	Nurse	Hospital	Rural
IDI11	29	F	4 years	Physician	Nursing home	Rural
IDI12	22	F	6 months	Nurse	Hospital	Urban
IDI13	27	F	4 years	Nurse	Hospital	Rural
IDI14	28	F	5 years	Psychologist	Nursing home	Urban, Rural
NFG1	39	F	15 years	Nurse	Hospital	Urban
NFG2	26	F	6 months	Nurse	Hospital	Urban
NFG3	44	F	20 years	Nurse	Hospital	Urban, Rural
NFG4	40	M	20 years	Nurse	Hospital	Urban, Rural
NFG5	29	F	1 year	Nurse	Nursing home	Rural
NFG6	21	F	6 months	Nurse	Hospital	Rural
IFG1	22	F	6 months	Nurse	Primary care	Urban
IFG2	22	F	6 months	Psychologist	Nursing home	Rural
IFG3	40	F	18 years	Physician	Primary care	Rural
IFG4	28	F	4 years	Physician	Hospital	Urban
IFG5	31	F	8 years	Nurse	Nursing home	Urban
IFG6	24	M	1 year	Psychologist	Nursing home	Urban

Note. IFG: Interdisciplinary Focus Group. NFG: Nurses Focus Group. IDI: In-depth Interview. F: female. M: male.

**Table 2 ijerph-18-12071-t002:** Interview protocol.

Stage	Subject	Content/Example Questions
Introduction	Motives	Healthcare professionals’ perceptions of loneliness in adults offers a lesson for all.
	Intentions	To carry out research to find out these perceptions.
	Ethical issues	Inform about the voluntary nature of participation, consent, possibility not to respond, to withdraw from the study at any point and confidentiality.
Beginning	Introductory question	“Please, tell me about how you perceive loneliness amongst older people living in the community and in long-term care settings.”
Development	Conversation guide	“How do the older adults usually express this feeling to you? Tell me about the barriers older adults may encounter when trying to interact with their relatives or other care home residents? How do you think healthcare professionals can contribute to fighting loneliness?”
Closing	Final question	“Is there anything else you would like to add?”
	Appreciation	“Thank you for your time and attention. Your participation will be very useful.”
	Offering	“We would like to remind you that you can contact us with any further questions. When we have the results of the study, we will inform you.”

**Table 3 ijerph-18-12071-t003:** Themes, subthemes and units of meaning.

Themes	Subthemes	Units of Meaning
When one’s personal life and social context lead to loneliness	Age, physical condition and character. Individual obstacles limiting accompaniment	Neglect, mobility problems, not wanting to be a nuisance, postmortem fidelity and preference
	Social factors that push older adults towards loneliness	Remoteness, economic inequality, getting rid of the elderly, work-related reasons, rural isolation, out of the system and abandonment
From abandonment to personal growth: the two faces of loneliness	Loneliness as an opportunity: between introspection and flight	Looking better and not wanting to be a nuisance
	Loneliness as a source of negative feelings	Feeling lonely, feelings of guilt, fear of the unknown, abandonment and closed in oneself
Loneliness as a health issue that needs to be addressed	Loneliness as a health issue	Health, disorientation, need for communication, somatising loneliness, need for relationships, demand for attention, at home: alone and at home: no stimulation
	The need for a multidimensional solution	Financial support, limitations of professionals, day centre, at home: integrated, in the community, reintegrate into the system, professionals: improvements, accompanied: participatory, need for encouragement, professionals: actions and resources

## Data Availability

Data are available from the authors (I.D.-S., C.R.-G. and C.F.-S.).

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
