# Peer review of "Healthcare Professionals’ Perceptions of Loneliness amongst Older Adults: A Qualitative Study"

_ijerph, 2021, doi:10.3390/ijerph182212071_

Round 1

Reviewer 1 Report

Dear authors,

The paper entitled "Healthcare professionals’ perceptions of loneliness amongst older adults: A qualitative study" brings an interesting and innovative perspective on how health professionals perceive and can potentially detect loneliness in seniors. Little is said about this topic, and it is become increasing important. I found the paper very well prepared, I just leave some suggestions that I think may help, before publication:

The fact that there is a large discrepancy in the professions of health professionals can be seen as a limitation.

The introduction is succinct, but complete, and highlights the main points surrounding loneliness and the importance of health professionals in detecting and referring seniors who show loneliness.

Great that they used COREQ.

Young sample, 29 years old on average, which may affect the identification processes by the seniors, and later hinder the sharing of feelings of loneliness. This can also affect the way in which information about seniors' loneliness is transmitted to health professionals, and subsequently condition the perception of health professionals about seniors' loneliness.

The same protocol was used for focus groups and for individual interviews, which ends up creating congruence in data collection through these two routes. However, it is important not to forget that the information obtained is from a different nature. The information shared, disposition and interventions during focus groups may have been qualitatively influenced by the group. Unlike individual interviews. This is a limitation of the study.

So good and rigorous thematic analysis following the Braun and Clarke principles

I have nothing to add to the analysis, it is complete, clear and rich.

I think this study, despite its limitations, brings an important and innovative perspective. The way we identify seniors with loneliness will become increasingly important due to the global aging of the population and the digital phenomena that isolate less socially supported seniors with less digital literacy from some resources. In this sense, and due to the importance of this theme, I challenge the authors to build a small and last section in the discussion where they propose some suggestions based on the evidence found for health professionals (in this case more directed to nurses, due to the sample) in the identification, prevention and management of loneliness in seniors in the health context.

Reviewer 2 Report

GENERAL

Line 10. The superscript should say “3”.

ABSTRACT

Line 20. Spell out the number 3 as ‘three’.

Line 22. Put an ‘and’ before the ‘3)’ thus ‘…loneliness”; and 3) “loneliness as a health …’.

Line 25. What do you mean with the term ‘social implication’?

Line 25. Why is there a line through the word ‘against’?

KEYWORDS

Line 27. Why use ‘content analysis’ when it is only mentioned once? Why not mention it again in the body of the text? Also, you only use the term ‘older person’ once and then as a keyword.

Choose your keywords carefully to make sure to maximise the visibility of the article. Heaps of suggestions if you search “how to choose keywords for academic articles”.

INTRODUCTION

Lines 52-53. The sentence starting ‘However, there is a lack…’ should be backed up with a citation and maybe further arguments as to why this study is needed. The following sentence (lines 54-56) is very broad  and could maybe be expanded on further in relation to the benefits of getting a better understanding of healthcare professionals’’ perceptions of loneliness. Thus, surely more literature must exists on this matter to back up the need for this study.

Lines 56-58. Make sure not to repeat exactly a sentence from the abstract.

METHOD

Line 61. Here you say that you follow a ‘descriptive qualitative study design’ but in the abstract you say that a ‘descriptive qualitative study was designed’? Do both of these add up?

Line 61. Please provide a citation for the design in question.

Line 90. Put the version number of ATLAS.ti in the brackets or maybe put ‘ATLAS.ti (version 8)’.

RESULTS

Lines 151-153. Maybe put an empty line above and below quotes throughout?

Line 188. Remember the comma in “e.g.,” and “i.e.,” not “e.g.” or “i.e.” respectively.

DISCUSSION/LIMITATIONS/CONCLUSION

Make sure to focus on the findings described in your manuscript and not to extend your conclusions beyond your findings.

TABLES/FIGURES

Table 1:

  • Start the table note section with ‘Note. IFG: interdisciplinary…’

Table 1, 2, and 3:

  • Using centred text makes it hard to read the tables, especially Table 2.

Reviewer 3 Report

This was a qualitative study on healthcare professional's perception of loneliness among older adults in Spain. The study addressed an important public health issue and was well-conducted. The findings were well-presented. I only have minor suggestions below for the authors to consider to improve the manuscript:

1) Table 1 Socio-demographic data (p. 2): Would it be possible to provide race and/or ethnicity data of the participants either in the Table or in the texts (Section 2.2)?

2) Section 2.3, Data collection: The authors wrote that "the other FG was comprised of nurses with experience looking after lonely older populations" (p. 3, line 85).  Were these nurse participants indeed experienced in looking after lonely older adults, or were they experienced in looking after older adults in general? If the latter, suggest to delete "lonely" in the description.

3) Sections 2.3 and 2.4, Data collection and data analysis: Were the interviews conducted in Spanish? Were the transcripts prepared in Spanish and then translated to English (for inclusion in this manuscript)? Please provide details.

4) Results-- 2nd theme: Suggest to revise the heading from "From abandonment to personal growth. The two faces of loneliness" to "From abandonment to personal growth: the two faces of loneliness"

5) Results-- 3rd theme: Suggest to either use "health issue" or "health-related issue" consistently in the theme and sub-theme 3.3.1, and throughout the manuscript. I prefer "health issue" for succinctness, but the authors may choose either as they deem appropriate.

6) Results-- 3rd theme, 3.3.2: Suggest to revise the sub-theme from "The need for a multidimensional approach" to "The need for a multidimensional solution"

7) Discussion-- The authors rightfully highlighted that this study did not take into account the perception of the older adults themselves and that this was one of the limitations (p. 10, lines 386-7). Can they add a few sentences to elaborate the implications of this limitation, and what can be done about it in the current and/or future study?

8) There were occasional typos, grammatical errors, and problems in language use. Please fix them. Examples are as follows:

  • "Desing" should be "Design" (p.2).
  • "Although the patient themself is aware of the origin of his problem.." (p. 8, line 275)-- "themself" should be "himself".
  • "... the generalised nature of our conclusions" may be better presented as ".. the generalizability of our conclusions" (p. 10, line 381)
  • "... only in-depth interviews and of focus groups were..." (p.10, line 390)-- "of" should be deleted.
